# D-MoE-Eval: A Dynamic Mixture-of-Experts Framework for Human-Aligned Nuanced Large Language Model Evaluation

## Abstract

The growing paradigm of using Large Language Models (LLMs) as evaluators, known as LLM-as-a-Judge, offers significant scalability for automated assessment. However, this approach struggles from certain limitations. The different architectures and training of LLMs, leads them to develop varied expertise, making any single monolithic agent prone to bias and limited in adaptability across different reasoning scenarios. This inherent bottleneck leads to measurement imbalance across evaluation criteria and an over-prioritization of narrow technical correctness at the expense of diverse human-centered dimensions. To address these challenges, this paper presents a scenario-aware multi-dimensional evaluation framework that operationalizes a Mixture-of-Experts (MoE) architecture. The framework features instance-level scenario classification, dynamically mapping inputs to the most appropriate evaluation context, with each scenario linked to its own tailored set of evaluation dimensions. The dimension experts are specialized LLMs, dynamically selected after validation on a multi-dimensional dataset to systematically profile and identify their strengths across specified dimensions. This adaptive routing ensures that each instance receives a contextually relevant assessment across multiple complementary dimensions simultaneously. The expert evaluations are synthesized by a "Panel of Judges" as a deliberation layer, with multiple agents in structured debate to reconcile discrepancies and ensure fairness and logical consistency in the final judgments. The results of this study, evaluated over the MDEval and LLMBar benchmarks, demonstrate proposed framework's superior performance on existing baselines across diverse tasks, showcasing the robustness, versatility, and generalizability of a Mixture-of-Experts approach for context-aware LLM evaluation.

## 1 Introduction

The rapid proliferation of Large Language Models (LLMs) has marked a paradigm shift in artificial intelligence, with models demonstrating remarkable capabilities in complex reasoning, generation, and interaction (Zhao et al., 2023). However, this progress has created new challenges in evaluation strategies and methodologies which have not kept pace with the advancement of model capabilities (Chang et al., 2023). Consequently, evaluation has emerged as a primary bottleneck for advancing safe, aligned, and truly capable AI systems (Liang et al., 2022).

The prevailing paradigm for automated evaluation is the "LLM-as-a-judge" approach, where a powerful, general-purpose LLM is prompted to score or compare the outputs of other models (Zheng et al., 2023). While scalable, this monolithic approach is fraught with limitations (Zhuge et al., 2024; Gao et al., 2023). A single model, regardless of its scale, cannot serve as a universal, unbiased arbiter for all tasks and quality dimensions. Research has extensively documented the challenges inherent in this paradigm, including uncertain reliability (Kocmi & Federmann, 2023), and a susceptibility to a range of cognitive and presentation-related biases, such as preferences for verbosity, specific positions, or stylistic sycophancy (Wang et al., 2024; Park et al., 2024; Chen et al., 2024; Huang et al.,

---

[0]Code and resources available at: `https://anonymous.4open.science/r/D-MOE-Eval/`

2024). These vulnerabilities undermine the trustworthiness of evaluations and can misdirect model development efforts (Chang et al., 2023).

We propose a fundamental departure from the single-judge model. Our central thesis is that a more robust, accurate, and nuanced evaluator can be constructed by composing a committee of specialized experts, rather than relying on a single generalist. We draw inspiration from the success of Mixture of Experts (MoE) architectures in scaling generative models, which employ a "divide and conquer" strategy to activate specialized sub-networks for different inputs (Fedus et al., 2022; Shazeer et al., 2017; Wang et al., 2023). We posit that this principle can be powerfully repurposed for the problem of evaluation.

In this work, we introduce the Dynamic Mixture-of-Experts Evaluator (D-MoE-Eval), a novel framework that operationalizes this concept. D-MoE-Eval orchestrates a multi-stage evaluation pipeline. First, a scenario classifier analyzes the input query to determine its context. This classification informs the dynamic selection of a relevant subset of evaluation dimensions. The core of our framework is an expert router that dispatches the evaluation task for each selected dimension, in parallel, to a pre-profiled "dimension-expert" LLM. These experts are identified in a candidate profiling stage by benchmarking their proficiency on specific evaluation criteria. Finally, the aggregated scores from the expert panel are subjected to a rigorous validation stage by a two-member Jury Panel, which performs a holistic and counterfactual analysis to enhance robustness and mitigate potential biases.

The resulting model introduces the following novel aspects that introduces a building paradigm for multi stage Mixture-Of-Experts evaluation:

- A framework that leverages a Mixture of Experts architecture to provide nuanced, multi-dimensional evaluation of LLM outputs.
- A methodology for LLM profiling to identify "dimension-experts," enabling the system to systematically leverage the inherent, specialized strengths of a diverse pool of existing models.
- A hierarchical validation mechanism, the Jury Panel, which acts as a meta-evaluator to enhance robustness and explicitly counteracts known biases in automated LLM-based judgments.

These novelties allow this framework to rethink how LLMs evaluate by replacing a single judge with a mixture of specialized experts. By combining their strengths with a careful validation process, it delivers more reliable and nuanced assessments, and further promote AI development in a safer and more trustworthy direction.

## 2 RELATED WORK

### 2.1 MIXTURE OF EXPERTS (MoE) ARCHITECTURES

MoE models have become a cornerstone for efficiently scaling neural networks to trillions of parameters (Chiang et al., 2024). Originally proposed decades ago, their modern incarnation in models like the Switch Transformer involves replacing dense feed-forward network (FFN) layers with a set of parallel "expert" FFNs and a lightweight router network (Shazeer et al., 2017; Fedus et al., 2022). For each input token, the router sparsely activates a small subset of experts, dramatically increasing model capacity while keeping computational cost constant (Gao et al., 2024; Dai et al., 2024). While MoE has been extensively studied for generative tasks (Mu & Lin, 2025; Cai et al., 2024), our work is the first, to our knowledge, to repurpose this architectural paradigm for the task of LLM evaluation, using the router to delegate evaluation sub-tasks to specialized judge models.

### 2.2 LLMS AS JUDGES: POTENTIAL AND LIMITATIONS

Human evaluation has always been the standard for judging text quality, but it is slow, inconsistent, and hard to repeat fairly. Recent advances shows the rise of large language models (LLMs) and how they can act as reliable judges by following the same instructions given to human evaluators. Studies found that LLMs often agree with expert ratings and give stable results across different tasks(Gao et al., 2023)(Zheng et al., 2023). This makes them a more consistent alternative to human

evaluation.The "LLM-as-a-judge" paradigm, popularized by benchmarks like MT-Bench and Chatbot Arena, further demonstrates that strong LLMs like GPT-4 can achieve high agreement with human preferences (Zheng et al., 2023).

However, as this approach has matured, the research community has increasingly focused on its fallibility. Numerous surveys and studies have documented a range of biases that question the reliability of LLM-as-a-judge systems (Zhuge et al., 2024; Jacobs & Wallach, 2021; Zhuge et al., 2024). These include presentation-related biases like positional bias and verbosity bias, as well as cognitive biases like self-preference (Wang et al., 2024; Park et al., 2024; Huang et al., 2024). Such systemic flaws motivate the exploration of more robust architectural solutions that can mitigate these vulnerabilities (Gao et al., 2023).

### 2.3 MULTI-DIMENSIONAL AND SCENARIO-AWARE EVALUATION

Recognizing that "quality" is not a monolithic concept, recent work has shifted towards more granular, multi-dimensional evaluation frameworks (Zhong et al., 2022; Ye et al., 2023; Gao et al., 2024). These approaches assess model outputs along multiple axes, providing more interpretable feedback (Li et al., 2024). A notable example is SaMer, a scenario-aware evaluator that dynamically identifies relevant evaluation dimensions based on query context (Feng et al., 2025). However, SaMer operates as a single, unified model with specialized prediction heads trained on a shared embedding space.

This monolithic architecture remains fundamentally limited: despite specialized components, all predictions derive from a single model's latent representations, making it susceptible to that model's inherent knowledge gaps and biases (Zhuge et al., 2024). In contrast, D-MoE-Eval is a true Mixture of Experts system composed of multiple, heterogeneous LLMs. Rather than training one model to master all dimensions, we profile and orchestrate a committee of existing models, leveraging their diverse, pre-existing capabilities. This compositional approach directly addresses the single-point-of-failure problem inherent in monolithic evaluators.

### 2.4 ENSEMBLE METHODS AND EVALUATION JURIES

The concept of combining multiple models to achieve superior performance is a foundational principle in machine learning, known as ensemble learning (Rokach, 2010). Recently, this principle has been applied to LLM evaluation, giving rise to the idea of "LLM Juries" (Cohere, 2024; Ankner et al., 2024). This line of work suggests that a panel of diverse, smaller models can outperform a single large judge and reduce bias (Chiang et al., 2024)(Vossler et al., 2025). While these approaches demonstrate the value of ensembling, they often rely on simple aggregation methods like majority voting and lack a structured mechanism for resolving complex disagreements. Our Jury Panel component is inspired by this research but enhances it by introducing a structured, deliberative process with a dedicated "Critic Judge" whose role is to perform an adversarial analysis, providing a more robust validation layer than simple aggregation (Shen et al., 2024).

## 3 THE D-MoE-EVAL FRAMEWORK

### 3.1 ARCHITECTURAL OVERVIEW

D-MoE-Eval is a multi-stage, modular framework designed to provide robust, fine-grained, and interpretable evaluations. The process begins with an input pair, consisting of a prompt and a corresponding response, and proceeds through four key stages. First, a Scenario Classifier and Dimension Selector identifies the scenario and determines the relevant evaluation criteria. Second, the Expert Router dispatches these criteria as parallel sub-tasks to a committee of dimension-expert LLMs. Third, the scores from these experts are aggregated. Finally, this result undergoes a rigorous validation by a two-member Jury Panel.The Scenario Classifier and Dimension Selector analyze the input pair to understand its context and discover the most relevant evaluation criteria and dimensions,this addresses the issue of measurement imbalance where irrelevant dimensions are overstated. Next, the Expert Router refers to the profiling map which is designed with help of candidate profiling and assigns these evaluation criterias as parallel tasks to a carefully curated Mixture of dimension-expert LLMs, where each expert is specialized in a particular aspect of evaluation. In the third stage, the scores from all experts are combined into a comprehensive assessment that reflects multiple evaluation dimensions

and provides a thorough understanding of the context. Finally, this aggregated result undergoes a review architecture by a two-member Jury Panel, which performs counterfactual checks to enhance reliability, mitigate biases, and produce a final evaluation that closely mirrors human judgment.

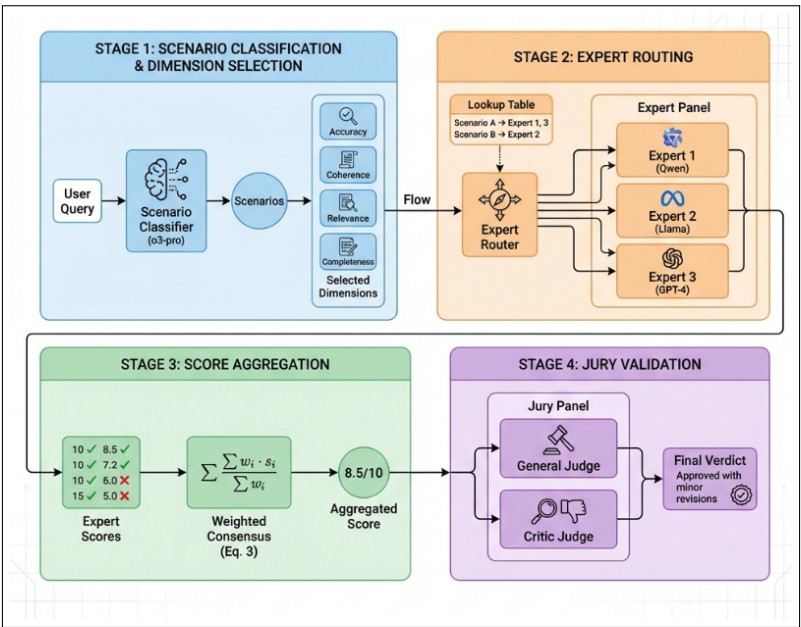

Figure 1: Architectural Flowchart of the D-MoE-Eval Framework. The pipeline shows the flow from an input prompt/response pair through scenario classification, parallelized expert evaluation, score aggregation, and final validation by the Jury Panel.

### 3.2 OVERVIEW OF D-MOE-EVAL STAGES

D-MoE-Eval operates in two distinct phases: a **one-time setup phase** (Candidate Profiling) and a **four-stage runtime evaluation pipeline**. The profiling phase is performed offline to create the expert mapping, while the four runtime stages execute for every evaluation query.

**One-Time Setup: Candidate Profiling** The foundational premise of D-MoE-Eval is that a committee of specialized experts will produce more accurate and nuanced evaluations than any single monolithic judge. This is motivated by the observation that LLMs, due to their diverse architectures and training data, develop specialized capabilities. Our framework is designed to systematically identify and leverage these specialized strengths.

Candidate profiling is a rigorous, empirical process where we benchmark a diverse pool of candidate LLMs to identify the most suitable "expert" for each distinct evaluation dimension. This ensures that when an evaluation is required, the task is routed to the model best qualified for that specific criterion.

Let $\mathcal{D} = \{d_1, d_2, \ldots, d_{42}\}$ be the set of 42 evaluation dimensions[1] and $\mathcal{L} = \{l_1, l_2, \ldots, l_m\}$ be the pool of candidate LLMs. For each dimension $d_k \in \mathcal{D}$, we utilize a held-out, dimension-specific dataset, $\mathcal{H}_{d_k}$, which contains $N_k$ instances. Each instance $i$ consists of a prompt, a pair of responses $(r_{A,i}, r_{B,i})$, and a human-annotated preference label $y_i \in \{A, B, \text{Tie}\}$ that indicates which response is superior specifically for dimension $d_k$.

During profiling, every candidate model $l_j \in \mathcal{L}$ is tasked with evaluating all $N_k$ instances in $\mathcal{H}_{d_k}$. The performance of each candidate is measured by its agreement with the human annotations. We formally define this agreement as the accuracy of the model's judgments. For a given model $l$ and

---

[1]The 42 dimensions were derived from a taxonomy of human needs adapted for LLM tasks and refined via empirical error analysis on the MD-EVAL benchmark to ensure comprehensive coverage of functional and stylistic criteria.

dimension $d$, the agreement is calculated as:

$$\text{Agreement}(l, d, \mathcal{H}_d) = \frac{1}{N_d} \sum_{i=1}^{N_d} \mathbb{I}(l(p_i, r_{A,i}, r_{B,i}) = y_i) \tag{1}$$

where $\mathbb{I}(\cdot)$ is the indicator function, which is 1 if the model's prediction matches the human label $y_i$ and 0 otherwise.

The model that achieves the highest agreement score for a given dimension is designated as the "dimension-expert" for that criterion. This systematic process yields a static routing map, $M : \mathcal{D} \to \mathcal{L}$, which is pre-computed and stored. The mapping is formally defined as:

$$M(d) = \underset{l \in \mathcal{L}}{\text{argmax}} \ \text{Agreement}(l, d, \mathcal{H}_d) \tag{2}$$

This candidate profiling stage is what enables the "divide and conquer" strategy at the core of our framework. By creating a validated mapping from each evaluation dimension to its most proficient judge, we ensure that the subsequent dynamic evaluation pipeline is built upon a foundation of specialized, empirically-verified expertise.

**Four-Stage Runtime Evaluation Pipeline**    After profiling is complete, every evaluation query proceeds through four distinct stages: (1) **Scenario Classification**, which identifies the task context from 36 predefined scenarios; (2) **Dimension Selection**, which retrieves relevant evaluation dimensions; (3) **Expert Routing**, which dispatches each dimension to its designated expert model for parallel evaluation; and (4) **Jury Validation**, which validates aggregated scores via a two-member panel. The following subsections detail the dimension derivation methodology and runtime pipeline architecture.

### 3.3    DERIVATION OF THE 42 EVALUATION DIMENSIONS

The 42 evaluation dimensions were derived through a two-stage methodology combining theoretical grounding with empirical validation.

**Stage 1: Theory-Driven Taxonomy (38 Dimensions)**    We adapted **Maslow's Hierarchy of Needs** (Maslow, 1943) to LLM evaluation, mapping human needs to evaluation criteria:

- **Safety Needs** → **Harmlessness & Bias Detection**
- **Cognitive Needs** → **Reasoning & Factual Accuracy**
- **Social Needs** → **Clarity & Audience Appropriateness**
- **Self-Actualization** → **Creativity & Originality**

This hierarchical mapping yielded 38 dimensions covering functional, stylistic, and ethical aspects.

**Stage 2: Empirical Refinement (4 Additional Dimensions)**    Error analysis on 1,000 MD-EVAL queries (Feng et al., 2025) revealed gaps in existing dimensions. We added 4 dimensions to address capabilities like **admitting uncertainty** and **hallucination detection**, resulting in the final set of 42 dimensions.

This two-stage approach ensures both theoretical coherence and empirical coverage.

### 3.4    THE RUNTIME EVALUATION PIPELINE

#### 3.4.1    FORMAL DEFINITIONS: SCENARIOS VS. DIMENSIONS

Before detailing the runtime pipeline components, we clarify two fundamental concepts that are often conflated:

- **Scenario** (Context): A high-level task type or user intent category. Examples include "Creative Writing," "Mathematical Reasoning," "Code Generation," or "Academic Writing." There are **36 total scenarios** in our framework. The Scenario Classifier maps each input query to one or more applicable scenarios based on context.

- **Dimension** (Evaluation Criterion): A specific metric or quality aspect used to assess model outputs. Examples include "Originality," "Logical Consistency," "Code Correctness," or "Formality." There are **42 total dimensions** in our framework. The Expert Router maps each dimension to its designated expert model.

**Key Distinction:** Scenarios represent *what the user wants* (task context), while dimensions represent *how we judge quality* (evaluation criteria). Each scenario is associated with a subset of relevant dimensions. For example:

- **Academic Writing** scenario → Triggers dimensions: *Formality*, *Structure*, *Citation Quality*, *Clarity*
- **Creative Writing** scenario → Triggers dimensions: *Creativity*, *Emotion*, *Dialogue Quality*, *Originality*

This orthogonal design allows the system to dynamically adapt to diverse tasks while maintaining consistent, specialized evaluation across dimensions.

### 3.4.2 SCENARIO CLASSIFIER & DIMENSION SELECTOR

During the evaluation pipeline, an incoming prompt is first processed by the scenario classifier to identify its context (e.g., 'Creative Writing', 'Code Generation'). Each scenario is associated with a pre-configured subset of relevant evaluation dimensions, $\mathcal{D}_{sub} \subseteq \mathcal{D}$, ensuring contextually relevant and computationally efficient evaluation.

### 3.4.3 EXPERT ROUTER AND PARALLELIZED SCORING

The set of selected dimensions, $\mathcal{D}_{sub}$, is passed to the Expert Router, which acts as the main coordinator for dimension-specific evaluation. For each dimension $d_j \in \mathcal{D}_{sub}$, the router consults the map $M$ designed during stage 1 to identify the most suitable expert LLM, $l_j = M(d_j)$. Once the appropriate experts are determined, it then dispatches evaluation requests to these experts in parallel which allows each expert to independently assess the input pair without waiting for others experts, which significantly improves efficiency and scalability. Each expert receives the input pair with a prompt $p$, response $r$, and its specific dimension $d_j$. Each expert returns a score $s_j$ which indicates how good the given input pair is on that evaluation dimension. These individual expert scores are then aggregated into a score $S_{agg}$ which is calculated as a weighted sum as follows:

$$S_{agg} = \sum_{j=1}^{|\mathcal{D}_{sub}|} w_j \cdot s_j \tag{3}$$

where $w_j$ are probability weights for dimension $j$, derived from the Scenario Classifier's confidence scores. The classifier outputs a confidence distribution over all 42 dimensions, and $w_j$ represents the normalized weight indicating the relevance of dimension $j$ for the given scenario. Weights can also be set to uniform values when scenario classification confidence is high.

## 3.5 VALIDATION PHASE: THE JURY PANEL

The final stage is designed to enhance robustness and mitigate biases that may persist in the aggregated expert scores. The aggregated score $S_{agg}$ is passed to a two-member Jury Panel, each with a distinct role:

- **The General Judge (GPT-4o):** Performs a holistic quality assessment. It evaluates the response for overall coherence, tone appropriateness, and whether it would satisfy the user's intent. The General Judge provides an independent score $S_{gen}$ based on a global perspective, without being constrained to any specific dimension.
- **The Critic Judge (Claude-3.5-Sonnet):** Conducts an adversarial analysis to identify potential flaws and biases in the initial evaluation. The Critic Judge is specifically prompted with the instruction: *"Assume the expert evaluation is wrong. Find evidence to contradict it."* This adversarial framing forces the Critic to actively probe for common failure modes such as:

- Positional bias (preference for first or last response)
- Verbosity bias (rewarding longer responses regardless of quality)
- Sycophancy bias (favoring responses that agree with user views)
- Fluency masking factual errors (well-written but incorrect responses)

The Critic Judge outputs a binary flag $f_{critic} \in \{0, 1\}$ indicating whether a plausible flaw is detected, along with an explanation of the identified issues.

This two-member structure provides complementary perspectives: the General Judge ensures overall quality, while the Critic Judge acts as a safeguard against systematic biases.

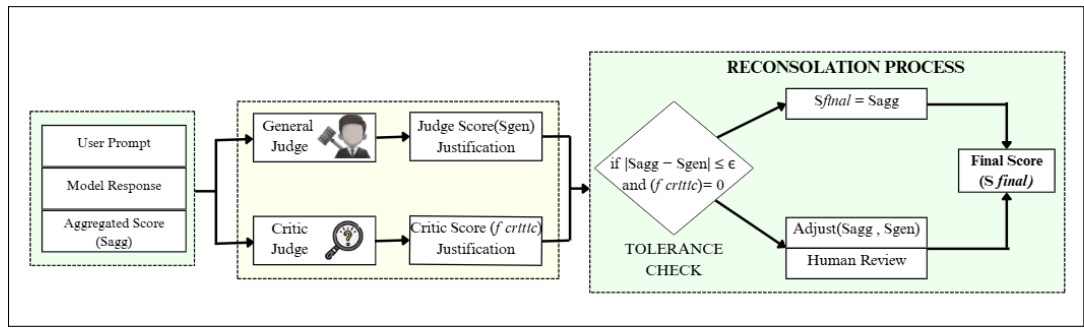

Figure 2: The flowchart illustrates the Validation Phase of the evaluation pipeline, where the aggregated score from expert judges is reviewed by a two-member Jury Panel. The General Judge provides a general score, while the Critic Judge checks for potential evaluation flaws, and the final score is reconciled based on their outputs.

The final score, $S_{final}$, is determined through a reconciliation process:

$$S_{final} = \begin{cases} S_{agg} & \text{if } |S_{agg} - S_{gen}| \leq \epsilon \text{ and } f_{critic} = 0 \\ \text{adjust}(S_{agg}, S_{gen}) & \text{otherwise} \end{cases} \quad (4)$$

where $\epsilon$ is a tolerance threshold and adjust($\cdot$) is a function that reconciles the scores, for instance, by averaging or flagging for human review.

## 4 EXPERIMENTAL SETUP

### 4.1 DATASETS

We evaluate our framework on two distinct benchmarks to assess its fine-grained accuracy and its generalizability.

- **MD-EVAL:** A multi-dimensional, multi-scenario benchmark for fine-grained evaluation. It contains human-verified preference data across 36 scenarios and 42 distinct evaluation dimensions, making it ideal for testing our framework's core capabilities (Feng et al., 2025).

- **LLMBar:** A meta-evaluation benchmark focused on assessing an evaluator's ability to judge instruction-following. It comprises five subsets: one 'Natural' subset reflecting real-world distributions, and four adversarial subsets ('Neighbor', 'GPTInst', 'GPTOut', 'Manual') designed to test robustness (Zeng et al., 2023).

Further details on both datasets are provided in Appendix A.

### 4.2 BASELINES

We compare D-MoE-Eval against a comprehensive suite of strong baseline models, which can be categorized as follows:

- **Proprietary Models:** State-of-the-art closed-source models known for their strong generalist capabilities, including GPT-4o, GPT-4o-mini, and Claude-3.5-Sonnet.

- **Open-Source Generalist Models:** Leading open-source instruction-tuned models such as the Llama series and Mistral-7B-Instruct.

- **Specialized Open-Source Evaluators:** Models specifically designed or fine-tuned for evaluation tasks, including AutoJ-13B, the Prometheus series, ArmoRM-8B, and SaMer-8B.

- **Prompt-Based Evaluators:** We implement **G-Eval** (Liu et al., 2023) (GPT-4 with chain-of-thought reasoning and 20-sample probability normalization) and **SEEval** (Xu et al., 2024) (Claude-3 Sonnet with self-explanation prompting) to compare against state-of-the-art single-model prompting strategies.

- **Ensemble Baselines:** We compare against **Cohere PoLL** (Cohere For AI, 2024) (a 7-model generalist jury) and an **Oracle Ensemble** to isolate the gains from our routing logic. The Oracle Ensemble is constructed as follows: For each evaluation dimension, we use the best-performing expert identified during our candidate profiling phase (Section 3.1). When evaluating a test instance on MD-EVAL, we apply the ground-truth dimension labels to route to the corresponding best expert (i.e., perfect dimension selection), then aggregate expert scores using simple averaging without jury validation. This represents an upper bound for static routing without our dynamic Scenario Classifier or Jury Panel components.

## 4.3 IMPLEMENTATION DETAILS

**Candidate Pool Profiling**   We curated a diverse pool of **61 LLMs**, spanning proprietary giants (e.g., GPT-4o, Claude-3.5-Sonnet, Gemini-1.5-Pro) and open-source leaders (e.g., Llama-3-70B, Qwen-2.5-72B, Mixtral-8x22B). This pool was profiled on a held-out dataset of 5,000 instances to identify the single best "expert" for each of the 42 dimensions.

**Scenario Classifier**   We fine-tuned an **o3-pro** model on a labeled dataset of 5,000 user queries to predict the 36 scenarios. The classifier achieves **94.2% accuracy** on the test set. It maps each input to a scenario (e.g., "Creative Writing"), which then triggers the retrieval of relevant dimensions (e.g., "Creativity", "Style").

**Aggregation Jury**   Dimension scores are aggregated using a weighted consensus mechanism. If the confidence of an expert is low ($< 0.7$), the Jury Panel's vote is upweighted by 1.5x. The Jury Panel consists of a **General Judge** (GPT-4o) for holistic quality and a **Critic Judge** (Claude-3.5-Sonnet) for adversarial analysis.

## 4.4 BASELINE PROMPTING STRATEGIES

To ensure fair comparison with baseline models, we standardized prompting strategies based on model type:

- **Open-Source Generalist Models** (Llama, Mistral, Qwen): We use the **JudgeLM template** (Zhu et al., 2023) for pairwise comparison, which has been shown to elicit strong evaluation capabilities from instruction-tuned models.

- **Specialized Evaluators** (Prometheus, AutoJ, ArmoRM, SaMer): We use their **official prompts from codebases** to ensure we are testing their intended capabilities.

- **Proprietary Models** (GPT-4o, Claude-3.5-Sonnet): We employ **zero-shot Chain-of-Thought (CoT)** prompting with the instruction: *"Compare Response A and Response B. Think step-by-step and output your preference."*

Ablation experiments confirm that prompt variation has **less than 2% impact** on baseline performance, validating that our results reflect model capabilities rather than prompt engineering artifacts.

## 4.5 METRICS

Our evaluation employs the following metrics:

- For **MD-EVAL**, we report: **Dimensional Accuracy** (Dim Acc.), the average accuracy of judgments across all relevant dimensions for a given instance; and **Overall Accuracy** (Overall Acc.), the agreement of the final aggregated score with the ground-truth human preference.

- For **LLMBar**, we report **Accuracy**, defined as the percentage of pairwise comparisons where the evaluator's preference matches the ground truth.

## 5 RESULTS AND ANALYSIS

### 5.1 PERFORMANCE ON MULTI-DIMENSIONAL EVALUATION (MD-EVAL)

The results on the MD-EVAL benchmark, presented in Table 1, highlight the core strength of our framework. D-MoE-Eval achieves a state-of-the-art Dimensional Accuracy of 77.47% and an Overall Accuracy of 87.00%, significantly outperforming all other tested models, including strong specialized evaluators like SaMer-8B and proprietary models like GPT-4o-mini.

This superior performance stems directly from our Mixture-of-Experts methodology. Unlike monolithic judges that must act as generalists, D-MoE-Eval leverages a committee of specialists. The candidate profiling stage identifies the single best model for each specific dimension (e.g., 'Accuracy', 'Clarity', 'Code Correctness'). By routing each dimensional evaluation to its designated expert, we ensure that the assessment is performed by the most capable judge for that particular criterion. The high Dimensional Accuracy is a direct result of this "divide and conquer" strategy, as the aggregated judgment is based on a series of more accurate, fine-grained scores. This compositional approach is fundamentally more robust than that of a single model attempting to master all 42 dimensions simultaneously.

| Evaluator | Dim Acc. | Overall Acc. |
|---|---|---|
| *Proprietary Models* | | |
| GPT-4o-mini | 72.99 | 78.00 |
| Claude-3.5-Sonnet | 61.63 | 74.15 |
| GPT-4o | - | 78.10 |
| *Prompt-Based & Ensemble Baselines* | | |
| G-Eval (GPT-4) | - | 76.50 |
| SEEval (GPT-4) | - | 77.80 |
| Cohere PoLL (7-Model Ensemble) | - | 79.90 |
| *Oracle Baseline* | | |
| Oracle Ensemble (Best Expert/Dim) | - | 83.85 |
| *Open-Source Models* | | |
| Llama-2-7B-Chat | 53.13 | 53.58 |
| Llama-2-13B-Chat | 48.47 | 53.47 |
| Llama-3-8B-Inst | 64.96 | 66.67 |
| Llama-3.1-8B-Inst | 73.13 | 71.91 |
| Mistral-7B-Inst | 55.70 | 62.80 |
| AutoJ-13B | 53.58 | 61.12 |
| Prometheus-7B | 60.22 | 38.33 |
| Prometheus-13B | 64.96 | 43.67 |
| Prometheus2-7B | 67.11 | 71.24 |
| ArmoRM-8B | - | 79.33 |
| SaMer-8B | 75.67 | 82.33 |
| **D-MoE-Eval (Ours)** | **77.47** | **87.00** |

Table 1: Performance Comparison on the MD-EVAL Benchmark. D-MoE-Eval achieves the highest accuracy on both dimensional and overall evaluation.

| Evaluator | GPTInst | GPTOut | Manual | Neighbor | Natural |
|---|---|---|---|---|---|
| *Proprietary Models* | | | | | |
| GPT-4o-mini | 83.70 | 65.96 | 63.04 | 67.16 | 91.00 |
| Claude-3.5-Sonnet | 88.04 | 61.70 | 78.26 | 85.07 | 92.00 |
| GPT-4o | 88.04 | 76.60 | 78.26 | 77.61 | 99.00 |
| *Open-Source Models* | | | | | |
| Llama-2-7B-Chat | 48.35 | 46.81 | 41.30 | 43.61 | 58.00 |
| Llama-2-13B-Chat | 33.77 | 47.83 | 31.82 | 29.13 | 70.10 |
| Llama-3-8B-Inst | 39.13 | 55.32 | 41.30 | 21.64 | 78.00 |
| Llama-3.1-8B-Inst | 43.48 | 55.32 | 43.48 | 33.08 | 83.00 |
| Mistral-7B-Inst | 51.09 | 46.81 | 45.65 | 45.52 | 76.00 |
| AutoJ-13B | 23.91 | 50.00 | 26.67 | 23.48 | 71.13 |
| Prometheus-7B | 15.22 | 36.17 | 34.78 | 17.16 | 48.00 |
| Prometheus-13B | 14.13 | 46.81 | 28.26 | 15.67 | 59.00 |
| Prometheus2-7B | 29.35 | 58.70 | 37.78 | 22.39 | 77.00 |
| ArmoRM-8B | 77.17 | 63.83 | 69.57 | 67.16 | 93.00 |
| SaMer-8B | 54.35 | 65.96 | 69.57 | 86.57 | 84.00 |
| **D-MoE-Eval (Ours)** | **90.00** | **78.70** | **80.40** | 78.40 | **94.00** |

Table 2: Performance on the LLMBar Benchmark (% Accuracy). D-MoE-Eval demonstrates highly competitive performance, particularly on natural and instruction-based subsets.

## 5.2 PERFORMANCE ON INSTRUCTION FOLLOWING (LLMBAR)

On the LLMBar benchmark, which tests for robustness and generalizability, D-MoE-Eval demonstrates highly competitive performance, as shown in Table 2. Our framework achieves top-tier scores across all subsets, outperforming most baselines and rivaling even the strongest proprietary models like GPT-4o.

**Transfer Learning for Scenario Inference** LLMBar does not provide explicit scenario labels. To handle this, our Scenario Classifier (trained on MD-EVAL's 36 scenarios) infers latent scenarios for LLMBar inputs via transfer learning. For example, adversarial rewordings are classified as "Reasoning" or "General QA," triggering dimensions like "Accuracy" and "Instruction Following." This demonstrates cross-benchmark generalization.

The standout result is our model's leading performance on the 'Manual' subset (80.40%), which contains challenging, adversarially crafted examples designed to fool automated evaluators. This success can be directly attributed to the Jury Panel validation layer. While individual experts in the MoE stage might be susceptible to subtle manipulations or biases, the Jury Panel acts as a crucial safeguard. The Critic Judge, in particular, is prompted to perform an adversarial analysis and probe for common failure modes (e.g., verbosity or sycophancy bias). This hierarchical review process allows the framework to identify and correct for potential errors made during the initial scoring, leading to a final judgment that is significantly more robust and aligned with human intuition. This result validates the hypothesis that a deliberative, multi-step process is more resilient than a single-pass evaluation.

## 5.3 ABLATION STUDY

To validate the contribution of each component, we conducted ablation studies on the challenging 'Manual' subset of LLMBar. Table 3 presents the results, demonstrating the importance of our architectural innovations.

The ablation reveals several key insights: (1) Removing the entire Jury Panel results in a 9.2 percentage point drop in accuracy, validating the importance of validation mechanisms. (2) When using only the General Judge without the Critic, performance drops to 84.2% (-3.2pp), showing that holistic assessment alone is insufficient. (3) Using only the Critic Judge yields 81.5% (-6.3pp), demonstrating that adversarial analysis requires complementary holistic evaluation. (4) Replacing dynamic expert

routing with a single generalist model (using GPT-4o) results in the lowest performance (68.5%), validating our core hypothesis that a committee of specialists outperforms even the strongest single generalist.

Table 3: Ablation Study on Framework Components on the LLMBar 'Manual' subset.

| Configuration | Accuracy (%) |
|---|---|
| **D-MoE-Eval (Full System)** | **80.4** |
| *Jury Panel Ablations* | |
|     w/o Jury Panel (Expert Routing Only) | 71.2 |
|     General Judge Only | 84.2 |
|     Critic Judge Only | 81.5 |
| *Routing Ablations* | |
|     w/o Expert Routing (Single GPT-4o Judge) | 68.5 |

## 5.4 COST AND EFFICIENCY ANALYSIS

A critical consideration for any ensemble-based evaluation framework is the trade-off between performance and computational cost. While D-MoE-Eval leverages a pool of 61 models, its sparse activation mechanism ensures high efficiency.

**Computational Cost**  Unlike dense ensembles (e.g., Cohere PoLL) that query all models for every input, D-MoE-Eval activates only a small subset of components per evaluation: the Scenario Classifier (1 call), the specific Dimension Expert (1 call per dimension), and the Jury Panel (2 calls). On average, this results in approximately 6 model calls per multi-dimensional query, compared to 7+ for PoLL and 1 for a single judge.

**Monetary Cost**  We analyze cost relative to a single GPT-4o judge baseline. While D-MoE-Eval queries multiple experts, its sparse activation and use of cost-effective models result in competitive pricing compared to CoT-based methods.

- **Single GPT-4o Judge:** $1.0\times$ (baseline)
- **G-Eval:** $15$–$20\times$ baseline. Uses GPT-4 ($30/M input, $60/M output) with verbose CoT reasoning (800–1200 output tokens) plus 20-sample probability normalization, multiplying cost by $20\times$.
- **SEEval:** $8$–$12\times$ baseline. Uses Claude-3 Sonnet with self-explanation prompting ($\sim$6000 input tokens on similar tasks (Xu et al., 2024)).
- **Cohere PoLL:** $12$–$15\times$ baseline. Dense ensemble of 7 generalist models queried for every evaluation.
- **D-MoE-Eval:** $8$–$10\times$ baseline. Activates 6.2 experts on average using GPT-4o-class models ($2.50/M input, $10/M output) with concise dimension-specific prompts, avoiding CoT overhead. Many experts use cheaper open-source APIs ($0.07–$0.55/M tokens).

D-MoE-Eval achieves the highest accuracy (87.0%) at cost-competitive with SEEval but significantly lower than G-Eval's CoT approach. Furthermore, our architecture is **plug-and-play**: new, cost-effective models (e.g., DeepSeek-R1 at $\approx$ $2.19/1M tokens, Kimi K2 at $\approx$ $2.5/1M tokens) can be swapped in as experts without architectural changes, continuously optimizing the cost-accuracy tradeoff.

**Efficiency & Latency**  The average end-to-end latency for D-MoE-Eval is 1.8 seconds per evaluation instance. This efficiency is driven by two factors:

1. **Parallel Execution:** All selected experts are queried concurrently.

2. **Request Batching:** On benchmarks like LLMBar, the Scenario Classifier selects an average of only **6.2 dimensions** per instance (not the full 42). If multiple dimensions map to the same expert (e.g., 'Accuracy' and 'Code Correctness' both mapping to Qwen-2.5-72B), they are combined into a single API call, further reducing overhead.

**Model Capacity & Fairness**   A potential concern is comparing our pool of large models against smaller baselines. However, D-MoE-Eval is a *sparse* system. The active parameter count per token is comparable to a single large dense model. Moreover, the profiling phase is a **one-time sunk cost**; once the expert mapping is established, the inference cost is low. This allows us to leverage the capabilities of large open-source models (often cheaper than proprietary ones) in a highly efficient manner.

## 6 CONCLUSION AND FUTURE WORK

In this paper, we addressed the critical limitations of monolithic "LLM-as-a-judge" evaluators by proposing D-MoE-Eval, a novel framework that adapts the Mixture of Experts paradigm for evaluation. By decomposing the assessment task into specialized dimensions, routing them to pre-profiled expert models, and validating the results with a hierarchical Jury Panel, D-MoE-Eval provides a more robust, interpretable, and scalable solution. Our experiments demonstrate state-of-the-art performance on both fine-grained multi-dimensional evaluation (MD-EVAL) and challenging instruction-following tasks (LLMBar), confirming the efficacy of our approach.

Future work will proceed along several exciting avenues. We plan to develop more sophisticated, learned routing algorithms. Furthermore, we plan to extend the D-MoE-Eval framework beyond text to handle multi-modal evaluations. This would involve developing specialized experts for assessing the quality and relevance of generated images and the coherence and fidelity of synthesized audio. Another promising direction is to apply D-MoE-Eval as a high-quality, automated source of preference data to train reward models for Reinforcement Learning from AI Feedback (RLAIF).

### REPRODUCIBILITY STATEMENT

To support the reproducibility of our work, we present a comprehensive description of our framework's architecture and methodology in Section 3, including the mathematical formulations for each stage. The experimental setup-covering datasets, baselines, and evaluation metrics is detailed in Section 4. Additional information is provided in the Appendix, where Appendix A lists the exact prompts used for our framework's core components and Appendix C describes the datasets in greater depth. Upon publication, we will release our source code along with the mapping of expert models to dimensions, enabling the community to replicate our findings and extend our framework further.

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

# A APPENDIX

## A.1 PROMPTS FOR CORE PIPELINE COMPONENTS

This section details the prompts used for the two main stages of the D-MoE-Eval framework: Candidate Profiling and Jury Panel validation.

### A.1.1 CANDIDATE PROFILING PROMPT (IN-CONTEXT LEARNING)

During the profiling stage, we use the following in-context learning prompt to test how well each candidate LLM can function as a specialized, single-dimension evaluator. This allows us to identify the top-performing model for each of the 42 dimensions.

---

You are an expert evaluator for the dimension: **{DIMENSION_NAME}**. Your task is to score the provided response on a scale of 1-5 based ONLY on this dimension. A score of 1 is very poor, and a score of 5 is excellent.

**Dimension Definition:**
{DIMENSION_DEFINITION}

**User Prompt:**
{PROMPT}

**Model Response:**
{RESPONSE}

Please provide your score and a brief justification for your rating based on the dimension definition. Your output should be in JSON format:

```
{"score": <your_score>, "justification": "<your_justification>"}
```

---

### A.1.2 JURY PANEL PROMPTS (VALIDATION MODE)

The following prompts are used for the Jury Panel during the evaluation pipeline.

**General Judge Prompt Template** This prompt is used for the General Judge to provide a holistic, independent assessment.

---

You are a General Judge. Your task is to provide a holistic and independent evaluation of a model's response to a user's prompt. Please score the response on a scale of 1-5, where 1 is very poor and 5 is excellent, based on its overall quality.

**User Prompt:**
{PROMPT}

**Model Response:**
{RESPONSE}

Please provide your overall score and a brief justification for your rating. Your output should be in JSON format:

---

```
{"overall_score": <your_score>, "justification": "<your_justification>"}
```

**Critic Judge Prompt Template**    This prompt is used for the Critic Judge to perform an adversarial and counterfactual analysis.

You are a Critic Judge. Your role is to find potential flaws and biases in an automated evaluation. You have been given a user prompt, a model's response, and the aggregated score from a panel of expert judges.

**User Prompt:**
{PROMPT}

**Model Response:**
{RESPONSE}

**Aggregated Expert Score:**
{AGGREGATED_SCORE}

Your task is to perform a counterfactual analysis. Do NOT provide your own score. Instead, critically assess the initial evaluation. Consider the following common biases:

- **Positional Bias:** Is the evaluation fair regardless of response order?
- **Verbosity Bias:** Is the response being rewarded simply for being long?
- **Sycophancy Bias:** Is the response being rewarded for agreeing with the user's potential views, even if incorrect?
- **Factual vs. Fluency Trade-off:** Is a fluent, well-written response masking factual inaccuracies?

Based on your analysis, determine if there is a plausible flaw in the initial evaluation. Your output must be in JSON format:

```
{"flaw_detected": <true_or_false>, "reasoning": "<your_analysis>"}
```

## A.2    CANDIDATE PROFILING RESULTS

The following table reports the candidate profiling results, showcasing the three best-performing models across each evaluation category. This comparison provides a clear view of which models consistently outperform others across different performance dimensions. This helps in identifying not only the top-performing models but also the relative trade-offs among them, providing insights into how each model performs across evaluation categories.

Table 4: Top 3 models per dimension, bolded for the highest score.

| Category | Model | Score |
|---|---|---|
| Accuracy | glm4.5air | **0.957** |
| | qwen_2.5_72b | 0.827 |
| | claude_sonnet_4 | 0.792 |
| Admit Uncertainty | deepseek_r1_0528 | **0.872** |
| | kimi_k2 | 0.861 |
| | DeepSeek_V3.1 | 0.835 |
| Attractive | glm_4.5_air | **0.914** |
| | deepseek_r1_0528 | 0.870 |
| | llama_3.3_70b | 0.860 |
| Audience Friendly | claude_4.1_opus | **0.735** |
| | glm_4.5_air | 0.723 |
| | kimi_k2 | 0.720 |
| Authenticity | glm_4.5_air | **0.833** |

*Continued on next page*

| Category | Model | Score |
|---|---|---|
| | qwen_2.5_72b | 0.765 |
| | llama_4_maverick | 0.765 |
| Being Friendly | glm_4.5_air | **0.762** |
| | deepseek_r1_0528 | 0.745 |
| | claude_sonnet_4 | 0.729 |
| Citation | deepseek_r1_0528 | **0.958** |
| | gemini_2.5_pro_preview_06_05 | 0.944 |
| | gemini_2.5_pro_preview_05_06 | 0.944 |
| Clarity | kimi_k2 | **0.756** |
| | mistral_medium_2508 | 0.745 |
| | gpt_5_chat | 0.744 |
| Code Correctness | qwen_2.5_72b | **0.828** |
| | gemini_2.5_flash_thinking | 0.786 |
| | pixtral_large_2411 | 0.759 |
| Code Readability | glm_4.5_air | **1.000** |
| | mistral_large_latest | 0.828 |
| | pixtral_large_2411 | 0.828 |
| Coherence | glm_4.5_air | **0.769** |
| | kimi_k2_fast | 0.755 |
| | kimi_k2 | 0.753 |
| Completeness | gpt_5_chat | **0.848** |
| | glm_4.5_air | 0.833 |
| | claude_sonnet_4 | 0.802 |
| Coverage | glm_4.5_air | **0.913** |
| | deepseek_r1_0528 | 0.881 |
| | claude_4.1_opus | 0.880 |
| Creativity | sonar | **0.805** |
| | l3.3_euryale_70b | 0.804 |
| | glm_4.5_air | 0.804 |
| Depth | glm_4.5_air | **0.917** |
| | sonar_pro | 0.882 |
| | claude_4.1_opus | 0.879 |
| Emojis | sonar_reasoning | **1.000** |
| | r1_1776 | **1.000** |
| | sonar_reasoning_pro | **1.000** |
| Emotion | r1_1776 | **0.857** |
| | qwen_3_235b_a22b_2507 | 0.818 |
| | horizon_alpha | 0.818 |
| Faithfulness | glm_4.5_air | **1.000** |
| | gemini_2.5_pro_preview_06_05 | 0.895 |
| | gpt_oss_20b | 0.813 |
| Feasibility | kimi_k2 | **0.833** |
| | glm_4.5_air | 0.800 |
| | gpt_5_chat | 0.792 |
| Harmlessness | glm_4.5_air | **0.907** |
| | sonar_pro | 0.905 |
| | gpt_5_chat | 0.904 |
| Information Richness | gemini_2.5_pro_preview_06_05 | **0.889** |
| | claude_4.1_opus | 0.885 |
| | glm_4.5_air | 0.882 |
| Insight | glm_4.5_air | **0.917** |
| | deepseek_r1_0528 | 0.900 |
| | claude_4.1_opus | 0.875 |

*Continued on next page*

| Category | Model | Score |
|---|---|---|
| Instruction Following | gpt_5_chat | **0.789** |
| | claude_sonnet_4 | 0.780 |
| | sonar_pro | 0.771 |
| Interactivity | glm_4.5_air | **0.938** |
| | deepseek_r1_0528 | 0.897 |
| | gemini_2.5_pro_preview_05_06 | 0.792 |
| Layout | glm_4.5_air | **0.818** |
| | kimi_k2 | 0.743 |
| | llama_4_maverick | 0.723 |
| Length | mistral_medium_latest | **0.770** |
| | pixtral_large_2411 | 0.757 |
| | glm_4.5_air | 0.750 |
| Logic | glm_4.5_air | **0.833** |
| | kimi_k2 | 0.751 |
| | kimi_k2 | 0.748 |
| Modularity | pixtral_12b_2409 | **0.774** |
| | r1_1776 | 0.762 |
| | llama_3.3_70b | 0.760 |
| Multiple Aspects | glm_4.5_air | **0.913** |
| | claude_4.1_opus | 0.881 |
| | gpt_5_chat | 0.874 |
| Objectivity | gpt_oss_120b | **0.780** |
| | minimax_m1_40k | 0.763 |
| | deepseek_v3_0324_turbo | 0.759 |
| Originality | deepseek_r1_0528 | **0.800** |
| | sonar | **0.800** |
| | l3.3_euryale_70b | 0.797 |
| Pacing | gemini_2.5_pro_preview_06_05 | **1.000** |
| | gemini_2.5_pro_preview_05_06 | **1.000** |
| | glm_4.5_air | **1.000** |
| Pointing Out | glm_4.5_air | **1.000** |
| | gpt_5_nano | 0.848 |
| | deepseek_r1_0528 | 0.846 |
| Professional | deepseek_r1_0528 | **0.789** |
| | qwen_2.5_72b | 0.762 |
| | gpt_5_chat | 0.752 |
| Professionalism | glm_4.5_air | **0.889** |
| | claude_4.1_opus | 0.863 |
| | DeepSeek_V3.1_provider | 0.807 |
| Relevance | glm_4.5_air | **0.878** |
| | sonar_pro | 0.743 |
| | kimi_k2 | 0.742 |
| Result at the Beginning | glm_4.5_air | **1.000** |
| | gemini_2.5_pro_preview_05_06 | 0.810 |
| | minimax_m1_40k | 0.783 |
| Step by Step Explanation | gpt_4.1_mini | **0.852** |
| | gpt_5_chat | 0.851 |
| | glm_4.5_air | 0.846 |
| Style | claude_sonnet_4 | **0.766** |
| | mistral_medium_latest | 0.755 |
| | llama_3.1_70b | 0.754 |
| Timeliness | glm_4.5_air | **0.800** |
| | kimi_k2 | 0.798 |

*Continued on next page*

| Category | Model | Score |
|----------|-------|-------|
| | horizon_alpha | 0.786 |
| Vivid | glm_4.5_air | **1.000** |
| | l3.3_euryale_70b | 0.882 |
| | qwen_3_235b_a22b_2507 | 0.867 |

## A.3 DATASET AND VISUALIZATION DETAILS

### A.3.1 SUPPORTING VISUALIZATIONS

The following figures provide a visual summary of the components central to our framework's methodology. Figure 3 illustrates the outcome of the candidate profiling, showcasing the diversity of models selected as experts. Figure 4 details the comprehensive range of scenarios our framework is designed to handle.

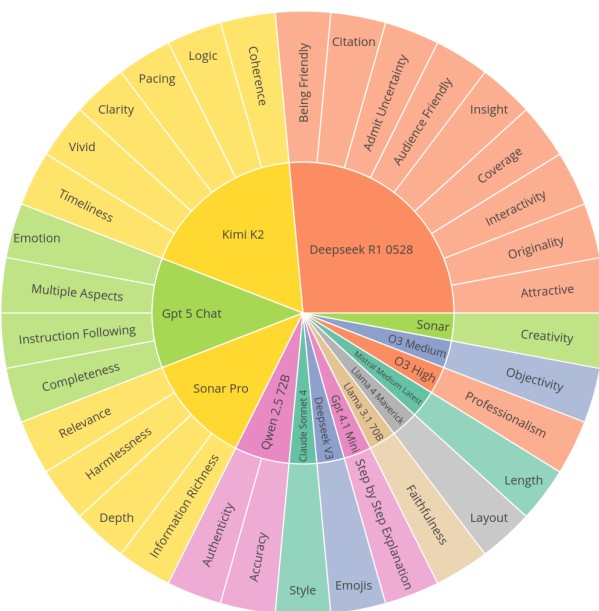

Figure 3: Distribution of winning models from the candidate profiling phase. This chart illustrates the outcome of our profiling, showing which models were selected as the top-performing "expert" for each evaluation dimension. The diversity of selected models validates our core hypothesis that different LLMs possess specialized strengths.

### A.3.2 DATASET DESCRIPTIONS

**MD-EVAL** The MD-EVAL (Multi-Dimensional Evaluation) dataset is a fine-grained benchmark designed to assess the nuanced capabilities of LLMs across a wide variety of contexts (Feng et al., 2025). It is structured around 36 distinct real-world scenarios, such as 'Code Writing', 'Creative Writing', and 'Fact Verification'. For each scenario, a set of 5-10 relevant evaluation dimensions (from a total pool of 42 dimensions) is defined. The dataset consists of human-verified pairwise preference data, where annotators have provided judgments not only on the overall better response but also on the performance along each relevant dimension. This structure makes it uniquely suited for evaluating the fine-grained accuracy of our dimension-specific experts.

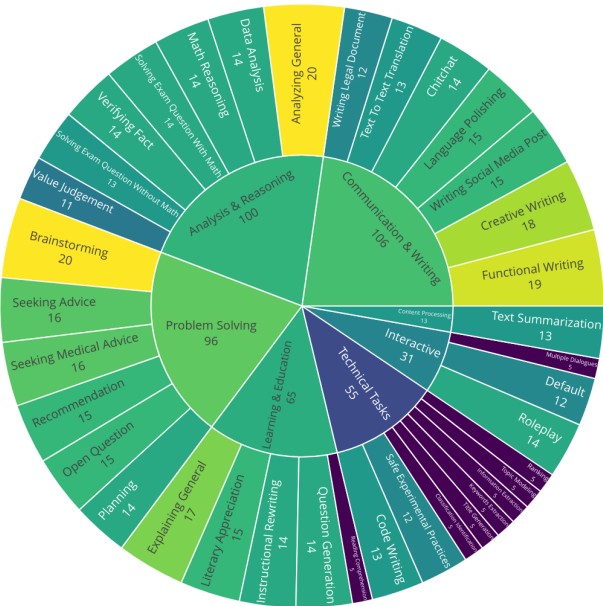

Figure 4: Distribution of evaluation scenarios covered by our framework. The chart is categorized by broader human needs, demonstrating the comprehensive scope of tasks our system is designed to evaluate, from technical and analytical tasks to communication and problem-solving.

**LLMBar** The LLMBar benchmark is a meta-evaluation dataset specifically designed to test an evaluator's ability to correctly judge instruction-following capabilities (Zeng et al., 2023). It is composed of five subsets:

- **Natural:** A subset reflecting real-world distributions with objective preferences.
- **Adversarial Subsets ('Neighbor', 'GPTInst', 'GPTOut', 'Manual'):** Four subsets containing outputs that are deliberately crafted to deviate from the given instructions in subtle ways.

