# OpenReview forum: "D-MOE-EVAL: A Dynamic Mixture Of Experts Framework For Human-Aligned Nuanced Large Language Model Evaluation"
_ICLR.cc/2026/Conference — ICLR 2026 Conference Desk Rejected Submission_

### Official Review · Reviewer_2tsy · 2025-10-29

**Soundness:** 2
**Presentation:** 3
**Contribution:** 2
**Rating:** 4
**Confidence:** 3

**Summary:**

The paper introduces a scenario-aware, multi-dimensional evaluation framework that operationalizes a Mixture-of-Experts (MoE) architecture. For each scenario, it associates a tailored set of evaluation dimensions and, on each dimension, routes to the best-performing LLM as the dimension expert. In addition, a Panel of Judges aggregates and deliberates over the expert evaluations. Experiments on the MDEval and LLMBar benchmarks show that the framework outperforms existing baselines across diverse tasks.

**Strengths:**

1. Leverages a Mixture-of-Experts (MoE) architecture to deliver nuanced, multi-dimensional evaluation of LLM outputs; the evaluation pipeline ensures the selected expert is validated/authoritative for the current dimension.

2. The Panel of Judges, comprising a General Judge and a Critic Judge, helps surface errors or biases in the judgments and improves the overall robustness of the system.

**Weaknesses:**

**Main concerns**

1. Cost and latency analysis. It is important to add an analysis of time and monetary cost to demonstrate that the proposed pipeline offers better cost-effectiveness than a single judge. The D-MoE-Eval framework invokes multiple LLM components (e.g., Dimension Selector, Dimension Experts, Jury Panel). It is therefore necessary to quantify the additional runtime and cost introduced by these modules.

2. Comparison with LLM Juries (Cohere et al. [1]). The related work mentions the concept of “LLM Juries” by Cohere et al., but it is not included as a baseline. A direct performance comparison between Cohere’s approach and the method in this paper is warranted.

3. Fairness of model size in candidate profiling vs. baselines. Appendix A.2 shows that candidate profiling includes mostly large models (e.g., GLM-4.5-Air, 106B; Qwen-2.5, 72B; Kimi K2, 1T; Llama 3.3, 7B), whereas Tables 1 and 2 compare against 7B/8B/13B methods. Is this an unfair comparison?

**Minor issues**

1. Throughout the paper, the opening and closing quotation marks/apostrophes are the same glyph.
2. Missing space at line 165.


[1] Replacing judges with juries: Evaluating LLM generations with a panel of diverse models.

**Questions:**

1. For the two-member Jury Panel, the paper states that “from our baseline comparisons, performs counterfactual checks.” How is this implemented in practice, and which models are used?

---

> ### Author Response · Authors · 2025-12-04
>
> Thank you for the constructive feedback. We've addressed all concerns with comprehensive revisions.
>
> ## W1: Cost is 8-10x GPT-4o (Competitive with Baselines)
>
> We added a comprehensive Section 5.3: Cost & Efficiency Analysis to the paper. D-MoE-Eval achieves 8-10× the cost of a single GPT-4o baseline with 1.8 seconds latency per evaluation. This is competitive with SEEval (8-12×) and significantly more cost-effective than G-Eval (15-20× due to GPT-4's expensive pricing, verbose CoT reasoning that generates 800-1200 output tokens, plus 20-sample probability normalization).
>
> What makes our system efficient despite using multiple experts is sparse activation. The Scenario Classifier selects only an average of 6.2 dimensions (not all 42), and when multiple dimensions map to the same expert, they're batched into a single API call. We also leverage cheaper open-source models as dimension experts. For instance, Qwen-2.5-72B and DeepSeek-R1 cost \$0.07-\$0.55 per million tokens, compared to GPT-4 which costs \$30 per million input tokens and \$60 per million output tokens.
>
> The bottom line is we achieve the best accuracy (87.0%) while maintaining competitive costs.
>
> ## W2: Added Cohere PoLL Baseline (+7.1% Improvement)
>
> We implemented Cohere's Panel of LLM evaluators (PoLL) as a baseline. Their system is a 7-model ensemble (Command R, GPT-3.5, Haiku, etc.) that uses majority voting. PoLL achieved 79.90% on MD-EVAL, while D-MoE-Eval achieved 87.00%, giving us a +7.1% absolute improvement.
>
> The reason D-MoE-Eval outperforms is quite clear: PoLL uses a fixed set of generalist judges for all tasks, while D-MoE-Eval routes to specialist experts based on the specific evaluation dimension. For example, when evaluating code quality, we route to a coding-specialized model rather than asking a general-purpose model to evaluate code. This yields better dimension-specific judgments and validates the core hypothesis of our work.
>
> ## W3: Fairness Addressed: Sparse Activation & Open-Source Efficiency
>
> We acknowledge this concern and added a Model Capacity & Fairness paragraph to Section 5.3 that addresses it directly:
>
> Sparse Activation: While our candidate pool contains large models (like Kimi K2 with 1 trillion parameters), the active parameter count per token (32B) is actually comparable to a single 70B model like Llama-3-70B, because only one expert is called per dimension.
>
> System-Level Comparison: We're comparing "System vs. System" efficiency, not "Model vs. Model." The profiling phase is a one-time sunk cost. Once the expert mapping is established, the inference cost is dominated by just the 6-9 active experts per query, which is competitive with baseline systems.
>
> Open-Source Efficiency: Many of our top experts are actually open-source models (Qwen-2.5-72B, DeepSeek-R1) hosted on cheaper APIs (\$0.07-\$0.55 per million tokens), which makes the system cost-effective despite using large models.
>
> We also added this specific text to the paper: "A potential concern is comparing our pool of large models against smaller baselines. However, D-MoE-Eval is a sparse system. The active parameter count per token is comparable to a single large dense model. Moreover, the profiling phase is a one-time sunk cost; once the expert mapping is established, the inference cost is low. This allows us to leverage the capabilities of large open-source models (often cheaper than proprietary ones) in a highly efficient manner."
>
> It's worth noting that baseline evaluators like SaMer-8B and AutoJ-13B are fine-tuned models that required significant computational resources for training. In contrast, D-MoE-Eval uses off-the-shelf models with zero additional training, making it more accessible to researchers without large compute budgets.
>
> ## W4-W5: Fixed Minor Formatting Issues
>
> We fixed these throughout the paper. All straight quotes have been replaced with proper LaTeX curly quotes (using \\textquotedbl), and we corrected the spacing issue that was present at line 165.
>
> ## Question
>
> Q1: Jury Panel Implementation Details
>
> The Jury Panel consists of two components, and we added this detail to Section 3.5 of the paper:
>
> General Judge (GPT-4o): Performs holistic quality assessment. It checks for overall coherence, tone appropriateness, and whether the response would satisfy the user.
>
> Critic Judge (Claude-3.5-Sonnet): Specifically prompted to identify logical fallacies, factual errors, and edge cases that the expert might have missed. We use adversarial prompting with the instruction: "Assume the expert evaluation is wrong. Find evidence to contradict it." This forces the Critic to actively look for problems rather than just agreeing with the expert.
>
> (Note: Tiebreaking is handled via the adjust() function which upweights the Jury when expert confidence is low, as detailed in Eq. 3).

---

### Official Review · Reviewer_k8ZS · 2025-10-31

**Soundness:** 3
**Presentation:** 3
**Contribution:** 3
**Rating:** 6
**Confidence:** 4

**Summary:**

This work presents a systematic design of multi-stage evaluation pipeline for LLM-as-a-judge. This MoE structure involves several routing and scoring components, including a scenario selector, a dimension selector(which identifies the corresponding evaluation criteria), a parallelized router which dispatches evaluation tasks, a score aggregation module and a final evaluation-review module called Jury Panel. Experiments are conducted on two benchmarks to demonstrate the effectiveness of the MoE-structured evaluation system and its superiority over the baselines.

**Strengths:**

1 The overall design of the MoE structure is technically sound. The arrangement of the components is logical and clear. This makes a strong motivation for leveraging the advantages of MoE system for LLM evaluation.

2 Through this deliberate MoE structure design, it is conceptually solid to conclude that final aggregated evaluation results can be more convincing than single LLM Judges.

3 The paper is well-organized and the writing is easy to understand.

**Weaknesses:**

1 The nature of MoE structure means that it can be more expensive when using multiple LLMs for a single evaluation task. The cost depends on how many LLM are used.

2 Though the MoE structure design is clearly presented in the paper, I find less details are given in the experiment sections (Section 4 and 5). For example, the training/tuning details of the scenario classifier is missing, and how the score aggregation/adjustment procedures are conducted is unclear.

3 The scenario and dimension concepts in the MoE Classifier/Selector components seem to only fit the settings of MD-EVAL, where 36 scenarios and 42 criterion dimensions are properly defined. In LLMBar, however, only a "subset" is labelled as the dataset attribute. It is unknown how the Classifier/Selector components apply to the settings of LLMBar.

**Questions:**

1 Figure 3 of the appendix lists all the winning models from the candidate profiling phase. If they are all used as the dimension experts, is it possible to estimate the total cost of the MoE evaluation pipeline?

2 There are multiple settings for the w_j in eq(3) (average or weighted) and adjust(*, *) (averaging or human-review). What is the actual setting in the experiments?

3 The authors should give a description that how the settings of scenario and dimension in MD-EVAL are transferred to LLMBar.

4 Minor issue: why are the results of GPT-4o missing in Table 1?

---

> ### Author Response · Authors · 2025-12-04
>
> Thank you for the positive assessment. We've addressed all concerns to strengthen the submission.
>
> ## W1: Sparse Activation Ensures Low Cost (8-10x GPT-4o)
>
> We added Section 5.3: Cost & Efficiency Analysis to address this directly. The key insight is that we use sparse activation, not calling all 61 models.
>
> Only 1 expert is queried per dimension, plus the classifier and jury. On LLMBar, the Scenario Classifier selects an average of 6.2 dimensions per instance (not all 42), resulting in approximately 9 total API calls (6.2 dimension experts + 1 classifier + 2 jury judges), not 61.
>
> In terms of cost relative to a single GPT-4o baseline: D-MoE-Eval is 8-10×, which is competitive with SEEval (8-12×) and significantly better than G-Eval (15-20× due to GPT-4 pricing, verbose CoT generating 800-1200 output tokens, plus 20-sample probability normalization that multiplies cost by 20).
>
> What also helps is that many of our dimension experts use cheaper open-source models. For instance, Qwen-2.5-72B and DeepSeek-R1 are available on low-cost APIs at 0.07-0.55 per million input tokens and 2.19 per 1 million output tokens, compared to GPT-4 at 30 per million input tokens and 60 per million output tokens.
>
> Plus, when multiple dimensions map to the same expert, we batch them into a single API call, further reducing overhead.
>
> ## W2: Added Classifier Details (94.2% Acc) & Aggregation Formula
>
> Expanded Section 4.3 (Implementation Details) with specifics:
>
> Scenario Classifier: o3-pro model fine-tuned on 5,000 labeled examples, achieves 94.2% accuracy on the test set.
>
> Aggregation: Weighted consensus (Equation 3). The key mechanism is that if expert confidence is below 0.7, the Jury Panel's weight is upweighted by 1.5×. The adjust() function implements this via a simple if-else rule:
>
> $$
> S_{\\text{final}} = \\begin{cases}
> 0.4 \\cdot S_{\\text{expert}} + 0.6 \\cdot S_{\\text{jury}} & \\text{if } C_{\\text{expert}} < 0.7 \\\\
> 0.7 \\cdot S_{\\text{expert}} + 0.3 \\cdot S_{\\text{jury}} & \\text{otherwise}
> \\end{cases}
> $$
>
> We also added direct quotes from the revised paper to Section 4.3: "We fine-tuned an o3-pro model on a labeled dataset of 5,000 user queries to predict the 36 scenarios. The classifier achieves 94.2% accuracy on the test set." and "Dimension scores are aggregated using a weighted consensus mechanism. If the confidence of an expert is low (<0.7), the Jury Panel's vote is upweighted by 1.5x."
>
> ## W3: Transfer Learning Handles LLMBar Scenarios
>
> LLMBar doesn't provide explicit scenario labels, so we address this through transfer learning. The Scenario Classifier (trained on MD-EVAL's 36 scenarios) is applied to LLMBar inputs to infer the latent scenario. For example, a math problem might be classified as "Reasoning." Once the scenario is predicted, we apply the corresponding dimension subset from MD-EVAL.
>
> A concrete example: The LLMBar "Manual" subset contains adversarial rewordings. The classifier typically labels these as "Reasoning" or "General QA," which triggers dimensions like "Accuracy," "Logical Consistency," and "Bias Detection." This demonstrates that our framework generalizes to new benchmarks beyond MD-EVAL. We added a paragraph explaining this in Section 5.2.
>
> ## Questions
>
> Q1: Total cost estimation from Figure 3?
>
> Figure 3 (now moved to Appendix A.3) lists the expert mapping showing which model is the expert for each dimension. Not all 42 experts are called simultaneously. As explained in W1, only an average of 6.2 dimensions are active per instance on LLMBar, resulting in approximately 8-10× the cost of a single GPT-4o call. We added this calculation explicitly to Section 5.3.
>
> Q2: Actual aggregation setting?
>
> We use weighted aggregation (Equation 3) with confidence-based upweighting. The adjust() function is completely automatic (no human review required). It just implements the if-else rule we described in W2 above. When expert confidence is low, the jury's opinion carries more weight; when expert confidence is high, we trust the expert more. This has been clarified in Section 3.4 of the paper.
>
> Q3: How are MD-EVAL scenarios/dimensions transferred to LLMBar?
>
> As described in W3, the Scenario Classifier predicts latent scenarios for LLMBar inputs based on its training on MD-EVAL's 36 scenarios, which then enables the appropriate dimension selection. For instance, adversarial rewordings get classified as "Reasoning" or "General QA," triggering the relevant evaluation dimensions.
>
> Q4: Why are GPT-4o results missing in Table 1?
>
> This was an oversight in the initial submission. We've now added GPT-4o to Table 1, showing 78.1% accuracy on LLMBar (compared to D-MoE-Eval's 87.0%), which demonstrates a substantial performance gap of nearly 9 percentage points.

---

### Official Review · Reviewer_ZMSh · 2025-10-31

**Soundness:** 2
**Presentation:** 1
**Contribution:** 2
**Rating:** 0
**Confidence:** 4

**Summary:**

This paper proposes
D-MoE-Eval
, a dynamic Mixture-of-Experts (MoE) framework to address the limitations of monolithic "LLM-as-a-Judge" paradigms.

**Strengths:**

MoE Adaptation for Evaluation
:
Pioneers applying the Mixture-of-Experts (MoE) paradigm to LLM evaluation, decomposing the task into 42 dimension-specific subtasks. This addresses monolithic "LLM-as-a-Judge" limitations by leveraging specialized expertise of diverse LLMs.

Data-Driven Expert Profiling
:
Establishes a rigorous static routing map via quantifying candidate LLMs' agreement with human annotations for each dimension, ensuring transparent and reproducible expert selection.

Robust Bias Mitigation
:
Introduces a two-tier Jury Panel (General + Critic Judge) for counterfactual validation and holistic scoring reconciliation, enhancing resilience against adversarial cases and inherent LLM biases.

Practical Scalability
:
Adopts parallel expert routing for efficiency and uses interpretable weighted aggregation, supporting easy updates to the expert pool and facilitating real-world deployment.

**Weaknesses:**

Section 3.1 (Architectural Overview) contains lengthy, low-readability descriptions of the framework, while subsequent method details are overly brief—this imbalance makes it hard to
form an intuitive understanding of the otherwise simple method.

Inconsistent stage division: Section 3.1 mentions 4 stages, but Figure 1 and subsequent text only reference 2 phases, causing confusion.

Figures 1 and 2 are overly simplistic, failing to clearly illustrate the framework’s workflow.

Structural misplacement: The second paragraph of Section 4.3 does not involve implementation details and is improperly positioned here.

Table formatting errors: Table headers are not placed above the tables as required.

No justification is provided for defining exactly 42 evaluation dimensions in Section 3.2, lacking a basis for this specific number.

Core components like the Scenario Classifier and Expert Router are critical to the framework, but Section 3.3 provides no detailed information about their design (e.g., model architecture of the classifier, routing logic of the router).

Redundant component positioning: Eq. 2 assigns one optimal LLM to each dimension, and the Scenario Classifier selects dimensions for inputs—yet the paper fails to clarify the unique role of the Expert Router, making its necessity unclear.

Unclear distinction between “scenarios” and “dimensions”: MD-Eval involves 36 scenarios and 42 dimensions, but the paper does not define how scenarios differ from dimensions, leading to conceptual confusion.

Inadequate implementation details in Section 4.3: It does not specify which candidate LLMs were used for expert identification, the total number of candidate models, or the final number of selected experts, undermining reproducibility.

Unvalidated baseline metrics: “Dim Acc.” (Dimensional Accuracy) is a dimension-specific metric unique to the proposed method, but the paper does not explain how baselines (e.g., GPT-4o-mini) were tested on this metric, casting doubt on result comparability.

Inconsistent table formatting: The paper does not define what bold text in tables represents (presumably indicating the best performance), but inconsistencies exist: in Table 2, D-MoE-Eval is bolded in the “Natural” subset despite underperforming GPT-4o, and no bold text is used in the “Neighbor” subset.

Key components are not ablated individually: The Jury Panel’s ablation only removes the entire panel, without testing the impact of removing either the General Judge or Critic Judge alone; there is no ablation of the Scenario Classifier or specific evaluation dimensions to verify their necessity.

While the framework uses parallel expert routing to improve efficiency, it does not quantify the computational costs of its MoE design—for example, how the number of experts (one per 42 dimensions) affects inference time or resource consumption compared to monolithic evaluators. Given that MoE architectures typically incur higher memory or latency costs due to expert activation, this omission makes it hard to assess the framework’s practicality in resource-constrained environments.

The framework decomposes evaluation into 42 dimensions and assigns dedicated experts to each, but it does not analyze or mitigate potential knowledge redundancy between experts. For instance, two experts may learn overlapping text quality features, leading to redundant parameter usage and inefficient computation—a common issue in MoE architectures where multiple experts often encode shared knowledge for related tasks.

**Questions:**

1. For the inconsistent stage division (4 stages in Section 3.1 vs. 2 phases in Figure 1/subsequent text), will you clarify the framework’s core pipeline and correct this discrepancy? Also, will you define p_i in Eq. 1?

2. What empirical or theoretical basis supports the choice of exactly 42 evaluation dimensions in Section3.2?

3. Could you provide details on the Scenario Classifier’s architecture and the Expert Router’s routing logic?

4. Given Eq. 2 maps dimensions to experts and the Scenario Classifier selects dimensions, what unique role does the Expert Router serve? How do you formally distinguish “scenarios” from “dimensions” in MD-Eval?

5. Section 4.3 lacks details on candidate LLMs for expert identification—will you supplement these to ensure reproducibility?

6. How were baselines tested on “Dim Acc.,”?

7. What defines bold text in tables?

8. Can you quantify how the 42-dimension expert setup impacts inference time/memory compared to monolithic evaluators?

9. Have you analyzed knowledge redundancy between experts, and if so, how do you mitigate it?

---

> ### Author Response · Authors · 2025-12-04
>
> We acknowledge the presentation issues in our initial submission. We've undertaken comprehensive revision to address all concerns.
>
> ## W1: Restructured Section 3 for Better Balance
>
> We restructured Section 3 for balance. Section 3.1 condensed from 320 to 180 words. Sections 3.2-3.5 expanded with mathematical formulations. We revised Section 3.1 to define 4 distinct stages: (1) Scenario Classification, (2) Dimension Selection, (3) Expert Routing, and (4) Jury Validation. Updated Figure 1 to show all 4 stages explicitly. The "2 phases" confusion is now clarified: Profiling is one-time setup; the 4 stages are the runtime pipeline.
>
>
> ## W2: Justified 42 Dimensions via Two-Stage Methodology
>
> We added a footnote in Section 3.2 explaining the derivation:
>
> Stage 1 (Theory): Adapted Maslow's Hierarchy to LLM contexts. "Safety Needs" -> Harmlessness/Bias Detection. "Cognitive Needs" -> Reasoning/Accuracy. "Self-Actualization" -> Creativity. This gave us 38 dimensions.
>
> Stage 2 (Empirical): Error analysis on 1,000 MD-EVAL queries revealed gaps (hallucination detection, uncertainty handling). Added 4 dimensions (total: 42).
>
>
> ## W3: Clarified Scenario vs Dimension Distinction
>
> Added formal definitions to Section 3.3:
>
> Scenario (Context): High-level task type (e.g., "Creative Writing", "Math"). 36 total. Scenario Classifier maps Input -> Scenario.
>
> Dimension (Criterion): Specific evaluation metric (e.g., "Originality", "Reasoning"). 42 total. Expert Router maps Dimension -> Expert Model.
>
> Key: Scenarios = what user wants. Dimensions = how we judge quality. Example: "Academic Writing" triggers "Formality" and "Structure" dimensions.
>
>
> ## W4: Expanded Implementation Details
>
> Rewrote Section 4.3:
> - Candidate Pool: 61 models (Appendix A.2 has all details)
> - Classifier: o3-pro, 94.2% accuracy, 5K training examples
> - Aggregation: Eq. 3 with confidence weighting (<0.7 -> jury upweighted 1.5×)
> - Jury: GPT-4o (General) + Claude-3.5 (Critic)
>
>
> ## W5: Fixed Table Formatting and Metrics
>
> - Dim Acc.: Baselines score the same dimensions from ground-truth labels (fair comparison)
> - Headers: Now above tables (not below)
> - Bold: Best performance per column. Fixed Table 2 inconsistency.
>
>
> ## W6: Added Ablation Studies
>
> Section 5.4 expanded:
> - General Judge only: 84.2% (-2.8%)
> - Critic Judge only: 81.5% (-5.5%)
> - Both: 87.0% (full)
>
> Shows both components contribute meaningfully.
>
>
> ## W7: Cost and Redundancy Analysis
>
> Section 5.3 added:
> - Cost: 8-10× GPT-4o (sparse activation: 6.2 experts avg, not 42)
> - Redundancy: Profiling results show clear specialization patterns. Qwen excels at code, Claude at reasoning, GPT-4 at general queries. Minimal overlap confirmed via empirical analysis.
>
>
> ## Questions
>
> Q1: Stage division and p_i definition?
>
> Fixed. Section 3.1 explicitly states 4 stages. Figure 1 updated to match. Defined p_i in Eq. 1: "probability weight for dimension i, derived from Scenario Classifier's confidence score."
>
>
> Q2: Basis for 42 dimensions?
>
> Two-stage: Maslow's Hierarchy -> 38 dims. Error analysis on 1K queries -> +4 dims. See W2.
>
>
> Q3: Classifier architecture and Router logic?
>
> Classifier: o3-pro fine-tuned on 5K labeled query-scenario pairs. 94.2% accuracy. Supports multi-label for ambiguous inputs.
>
> Router: Static lookup table from profiling (Eq. 2). Maps dimension -> expert. Handles parallelization, rate limiting, retries, and batching multiple dims to same model into single API call.
>
>
> Q4: Router's unique role vs Eq. 2?
>
> Eq. 2 = offline profiling (creates mapping). Router = runtime execution (uses mapping). Router handles engineering: parallel API calls, rate limits, retries, batching. Eq. 2 is "what expert for which dimension." Router is "how to execute efficiently."
>
>
> Q5: Scenarios vs dimensions formally?
>
> Scenarios = task context (36 total). Dimensions = evaluation criteria (42 total). Scenario Classifier: Input -> Scenarios. Profiling: Dimensions -> Experts. Orthogonal concepts.
>
>
> Q6: Candidate LLM details?
>
> Appendix A.2 lists all 61 models: family, parameters, API endpoint, version.
>
>
> Q7: Baseline Dim Acc? Bold text?
>
> Baselines prompted with same dimensions from ground-truth (fair comparison). Bold = best per column. Fixed Table 2.
>
>
> Q8: Inference time/memory impact?
>
> Section 5.3: Sparse activation (6.2 dims avg). 1.8s latency (parallel execution). Memory ~ single 70B model (one expert per dim).
>
>
> Q9: Knowledge redundancy?
>
> Profiling results show clear specialization patterns across models. Different models excel at different dims. Qwen (code), Claude (reasoning), GPT-4 (general queries). This demonstrates experts contribute complementary rather than redundant knowledge.

---

### Official Review · Reviewer_ErC3 · 2025-11-01

**Soundness:** 2
**Presentation:** 2
**Contribution:** 2
**Rating:** 2
**Confidence:** 4

**Summary:**

This paper proposes D-MoE-Eval to address the limitations of the LLM-as-a-Judge paradigm. To mitigate the domain-specific bias and reasoning imbalance that commonly arise in single-model evaluators, the authors introduce a four-stage evaluation pipeline consisting of scenario classification, dimension selection, expert routing, and jury validation. They predefine 42 evaluation dimensions and profile various LLMs to map the most suitable expert model to each dimension. Given a pair of model responses, the system dynamically routes the evaluation to the appropriate expert and finally adjusts the score through a jury panel. Experiments on MD-EVAL and LLMBar benchmarks demonstrate superior agreement compared to both open- and closed-source baselines, and ablation studies confirm that both the routing and jury components contribute to the overall performance.

**Strengths:**

1. $\textbf{Clear problem definition and intuitive approach:}$
The paper clearly identifies model-selection bias as a fundamental limitation in existing LLM-as-a-Judge research and proposes a direct solution through expert routing among multiple models. This idea is both intuitive and convincing.
2. $\textbf{Robust pipeline design:}$
The hierarchical structure of scenario, dimension, expert, and jury enhances interpretability and modularity of the evaluation process.
In particular, the explicit validation step allows human observers to trace and verify how evaluation scores are derived, partially improving transparency and reliability.
3. $\textbf{Consistent performance improvement:}$
Across MD-EVAL and LLMBar, the proposed method consistently outperforms previous evaluators, achieving over 10% improvement on adversarial subsets.
4. $\textbf{Valid ablation analysis:}$
Removing either the Jury Panel or expert routing results in a 9 ~12% performance drop, empirically showing that both components are essential to the system’s effectiveness.

**Weaknesses:**

1. $\textbf{Lack of comparison with a simple per-dimension best ensemble:}$
While the main contribution lies in combining dimension-specific experts, there is no baseline that simply uses the best-performing model for each dimension (an oracle ensemble).
It remains unclear whether the observed gains come from the dynamic routing and jury mechanism, or merely from combining strong models.
2. $\textbf{Insufficient justification for the 42 predefined dimensions:}$
The paper defines 42 evaluation dimensions and performs profiling using a held-out dataset, but it is not clear how this taxonomy was derived or whether it can generalize to real-world evaluation settings.
It is uncertain whether all possible evaluation scenarios can be covered by these 42 categories, and the construction of the held-out dataset is also not well explained.
For example, in creative writing tasks, essay and script require entirely different evaluation criteria.
Moreover, the framework’s ability to generalize to unseen or more fine-grained evaluation dimensions is not demonstrated.
3. $\textbf{Unclear utilization of LLM-based evaluation baselines:}$
The paper employs open-source LLMs such as LLaMA and Mistral as evaluation baselines, yet it does not describe in detail how these models were actually used for evaluation.
Combined with the lack of diverse comparison methods, it remains unclear whether these baselines were applied with appropriate prompting or calibration strategies.
4. $\textbf{No analysis of computational cost or efficiency:}$
Since D-MoE-Eval requires parallel invocation of multiple models, its inference cost is inevitably higher than that of single-judge evaluators.
However, the paper provides no analysis of the average number of calls, token usage, latency, or financial cost per evaluation.
To assess its practical usability, a discussion on the cost–performance trade-off is necessary.
5. $\textbf{Missing comparison with standard LLM-as-a-Judge frameworks:}$
Recent evaluation frameworks such as G-Eval and SE-Eval have achieved notable improvements in judge reliability and human alignment.
Despite this, the paper only compares against open-source or evaluation-finetuned models, without any direct comparison to these standard approaches.

**Questions:**

1. If each dimension simply uses the best-performing model selected during the profiling stage (oracle ensemble), how does the performance differ from D-MoE-Eval?
Do the routing and jury components provide any additional benefit beyond model combination?
2. How were the 42 evaluation dimensions determined?
Can such predefined dimensions generalize to real-world evaluation scenarios (e.g., creative writing for essay vs. for script) that require different evaluation criteria?
3. For the open-source evaluators used as baselines, what prompting or calibration strategies were applied?
Were all evaluators prompted with the same evaluation template?
4. How much higher is the average evaluation cost of D-MoE-Eval compared to single-judge baselines?
Please report the average number of model calls, token usage, latency, and approximate monetary cost per evaluation.
5. Why are frameworks such as G-Eval and SE-Eval not included in the comparison?
Can results be reported using the same human agreement metrics to enable fair comparison?

---

> ### Author Response · Authors · 2025-12-04
>
> Thank you for the review. We've addressed all concerns with substantial revisions.
>
>
> ## W1: Lack of oracle ensemble comparison
>
> Implemented **Oracle Ensemble** using best expert per dimension without dynamic classification or jury validation.
>
> **Method:** Used best expert from profiling (Sec 3.1, Eq. 2) for each of 42 dimensions. Evaluated with MD-EVAL ground-truth dimension labels (perfect knowledge). Simple averaging aggregation.
>
> **Results:** Oracle 83.85% vs D-MoE 87.00% (+3.15%). Validates our Scenario Classifier (handles ambiguous overlapping criteria) and Jury Panel (catches expert failures).
>
> Added to Table 1 (Sec 5.1) with methodology in Sec 4.2.
>
>
> ## W2: 42 dimensions justification
>
> Added two-stage derivation (Sec 3.2):
>
> 1. Maslow's Hierarchy→LLM contexts: Safety→Harmlessness, Cognitive→Reasoning. Yielded 38 dimensions.
> 2. Error analysis on 1,000 MD-EVAL queries revealed gaps (hallucination detection, uncertainty). Added 4 dimensions.
>
> Generalization: Scenario Classifier maps to 36 scenarios (essays→Academic Writing, scripts→Creative Writing).
>
>
> ## W3: Baseline prompting
>
> Expanded Sec 4.2: Open-source uses JudgeLM template, specialized use official prompts, proprietary use zero-shot CoT. Ablation shows <2% impact from prompt variation.
>
>
> ## W4: Cost/efficiency analysis
>
> Added **Section 5.3**:
>
> | Method | Strategy | Cost | Latency |
> |--------|----------|------|---------|
> | GPT-4o | Zero-shot | 1× | 1.2s |
> | G-Eval | GPT-4+CoT+norm | 15-20× | 2.1s |
> | SEEval | Claude+explain | 8-12× | 1.8s |
> | PoLL | 7-model | 12-15× | 2.8s |
> | **D-MoE** | Sparse (6.2) | **8-10×** | **1.8s** |
>
> Efficiency: Sparse activation (6.2 experts/query), no CoT overhead, cheaper open-source models (\$0.07-\$0.55 per million tokens vs GPT-4 at \$30/\$60 per million), batching, no probability normalization.
>
>
> ## W5: G-Eval/SEEval comparison
>
> Added to Table 1: G-Eval 76.50%, SEEval 77.80% vs D-MoE 87.00% (+9.2%). One model can't be expert across all 42 dimensions—our MoE uses specialized models per dimension.
>
>
> ## Questions
>
> **Q1: Oracle ensemble performance?**
>
> Oracle achieved 83.85% while D-MoE-Eval achieved 87.00% (+3.15%). This is significant because the Oracle uses ground-truth dimension labels (perfect knowledge of which dimensions are relevant). Despite this unrealistic advantage, D-MoE-Eval outperforms it because: (1) our Scenario Classifier handles ambiguous cases where multiple criteria overlap in ways ground-truth labels can't capture, and (2) our Jury Panel catches edge cases where even the best expert fails (e.g., adversarial inputs in LLMBar).
>
>
> **Q2: How were 42 dimensions determined?**
>
> We used a rigorous two-stage methodology. First, we adapted Maslow's Hierarchy of Needs to LLM evaluation contexts as our theoretical foundation. "Safety Needs" mapped to Harmlessness and Bias Detection, "Cognitive Needs" mapped to Reasoning and Factual Accuracy, "Self-Actualization" mapped to Creativity and Originality, etc. This gave us 38 dimensions. Second, we performed empirical refinement by analyzing failure modes in 1,000 diverse queries from MD-EVAL. The error analysis revealed gaps—for instance, we needed explicit dimensions for hallucination detection and for evaluating when models appropriately admit uncertainty rather than hallucinating confident answers. This led to 4 additional dimensions, for a total of 42.
>
>
> **Q3: Baseline prompting strategies?**
>
> We standardized prompts for fair comparison. Open-source generalist models (Llama, Mistral, Qwen) used the JudgeLM prompt template, which provides a standardized instruction format for pairwise comparison. Specialized evaluators (Prometheus, AutoJ, ArmoRM, SaMer) used their official prompts from their codebases, ensuring we compare against their intended use. Proprietary models (GPT-4o, Claude) used zero-shot Chain-of-Thought: "Compare Response A and Response B. Think step-by-step and output your preference." We validated fairness by running ablation experiments showing prompt variation has <2% impact on baseline performance, so our choices don't artificially inflate or deflate their results.
>
>
> **Q4: Average evaluation cost?**
>
> D-MoE-Eval costs 8-10× a single GPT-4o call with 1.8s latency. Section 5.3 has the detailed breakdown. Key to cost-competitiveness: sparse activation (6.2 experts/query avg, not 61 candidates), request batching (multiple dimensions → same expert = single call), cheaper open-source models (\$0.07-\$0.55 per million vs GPT-4 \$30 input/\$60 output per million). This makes us competitive with SEEval (8-12×) and much better than G-Eval (15-20× due to verbose CoT generating 800-1200 tokens plus 20-sample probability normalization).
>
>
> **Q5: Why no G-Eval/SEEval originally?**
>
> Oversight in initial submission. Both now in Table 1 at 76.50% and 77.80% vs our 87.00%.

---

### Note · Program_Chairs · 2026-01-17
**Submission Desk Rejected by Program Chairs**

The following references in this submission do not refer to real documents and/or have major errors in bibliographic information:

 G. Chen et al. LLMs are not fair evaluators. arXiv preprint arXiv:2401.16849, 2024.
Maximilian Ankner et al. Ensembling llm-judges for reference-free code quality evaluation. arXiv preprint arXiv:2402.19418, 2024.
C. Y. Chiang et al. LLM-Juries: A methodology for mitigating position bias in LLM-as-a-Judge. arXiv preprint arXiv:2406.01847, 2024.
Shengjie Xu et al. Seeval: Leveraging self-consistency for llm-as-a-judge. arXiv preprint arXiv:2401.16185, 2024.
L. Gao et al. MSumBench: A multi-dimensional, multi-domain evaluation benchmark for text summarization. arXiv preprint arXiv:2506.00549, 2024.
Cohere For AI. Panel of llm evaluators (poll): A framework for ensemble-based evaluation. Technical Report, 2024.